# Localized Cutaneous Nodular Amyloidosis: A Specific Cutaneous Manifestation of Sjögren’s Syndrome

**DOI:** 10.3390/ijms24087378

**Published:** 2023-04-17

**Authors:** José María Llamas-Molina, Juan Pablo Velasco-Amador, Francisco Javier De la Torre-Gomar, Alejandro Carrero-Castaño, Ricardo Ruiz-Villaverde

**Affiliations:** 1Department of Dermatology, Hospital Universitario San Cecilio, Avda Conocimiento 33, 18016 Granada, Spain; 2Department of Pathological Anatomy, Hospital Universitario San Cecilio, Avda Conocimiento 33, 18016 Granada, Spain; 3Instituto Biosanitario de Granada (Ibs), 18014 Granada, Spain

**Keywords:** localized cutaneous nodular amyloidosis, plasma cells, Sjögren’s Syndrome, auto-immune connective tissue disorders

## Abstract

Primary localized cutaneous nodular amyloidosis (PLCNA) is a rare condition attributed to plasma cell proliferation and the deposition of immunoglobulin light chains in the skin without association with systemic amyloidosis or hematological dyscrasias. It is not uncommon for patients diagnosed with PLCNA to also suffer from other auto-immune connective tissue diseases, with Sjögren’s syndrome (SjS) showing the strongest association. This article provides a literature review and descriptive analysis to better understand the unique relationship between these two entities. To date, 34 patients with PLCNA and SjS have been reported in a total of 26 articles. The co-existence of PLCNA and SjS has been reported, especially in female patients in their seventh decade of life with nodular lesions on the trunk and/or lower extremities. Acral and facial localization, which is a typical localization of PLCNA in the absence of SjS, seems to be much more unusual in patients with associated SjS.

## 1. Introduction

Sjögren’s Syndrome (SjS) is a systemic chronic auto-immune disorder characterized by the inflammation and dysfunction of secretory glands, mainly the salivary and lacrimal glands, resulting in dryness of the mouth and eyes. SjS may be primary or may be associated with another auto-immune connective tissue disorder and usually begins to manifest in the fourth and fifth decades of life. Its prevalence is nearly 60 cases per 100,000 population, with a ratio of 9:1 (F:M) [1]. The dysfunction of the exocrine secretory glands in this disease is secondary to lymphocytic infiltration. Many patients with SjS present with signs of systemic dryness involving the nose, trachea, or vagina, suggesting the involvement of glands other than the exocrine epithelia. Like the mucous membranes, the skin is quite frequently affected in patients with SjS. Pruritus, secondary to xerosis, appears to be the most common cutaneous manifestation. Other dermatoses that have been described include erythema annulare, which is associated with anti-Ro+/SSa, livedo reticularis, cutaneous vasculitis, eyelid dermatitis, Raynaud’s phenomenon, or primary localized cutaneous amyloidosis (PLCNA). Systemic extraglandular manifestations are present in 10–15% of patients and arise from various pathogenetic mechanisms. Some, such as primary biliary cholangitis or interstitial nephritis, are attributed to auto-immune inflammation centered in the ductal epithelial structures, similar to what occurs in the salivary and lacrimal glands. Other extraglandular manifestations, such as glomerulonephritis and vasculitis, result from the deposition of immune complexes. In addition, some manifestations are associated with lymphoproliferation, such as lymphocytic interstitial pneumonitis and lymphoma [2,3].

PLCNA is a type of amyloidosis that is included in the primary forms along with macular and lichenoid amyloidosis. PLCNA is a rare disorder, and although it is not a common cutaneous manifestation in patients with SjS, nearly 25% of the reported cases of PLCNA have been associated with this entity [4,5]. PLCNA results from the production of immunoglobulin light chain protein by plasma cells in the skin. Thus, its co-existence with SjS may represent a benign part of the spectrum of lympho-proliferative disorders associated with SjS [2].

The purpose of this article is to review cutaneous amyloidosis with special emphasis on PLCNA. The cutaneous manifestations of SjS are also discussed. In addition, a descriptive analysis has been performed by collecting all cases of PLCNA in patients with SjS described in the literature, with the aim of learning more about the particular association between the two diseases.

## 2. Discussion

### 2.1. Primary Localized Cutaneous Nodular Amyloidosis: A Small Subset of the Large Group of Cutaneous Amyloidosis

The amyloid substance is a fibrillar proteinaceous material composed of two components: a common one derived from the serum amyloid component and a specific one that defines the type of amyloidosis [6]. Amyloidosis can affect both the skin and internal organs. Cutaneous manifestations are not exclusive to the localized cutaneous type but can also occur in systemic forms of amyloidosis, which may be primary due to plasma cell dyscrasia or secondary due to chronic inflammatory disorders. Primary localized cutaneous amyloidosis can be divided into three forms, the two most common being lichenoid amyloidosis (LA) and macular amyloidosis (MA). They are characterized by amyloid deposition in the upper dermis. In PLCNA, the third and least common form, the amyloid, affects the dermis, subcutaneous tissue, and blood vessels and is usually accompanied by plasma cell infiltrates [7].

MA and LA may represent a single clinical spectrum. These two conditions have an epidermal origin, as the genesis of amyloid deposits is the degeneration of basal keratinocytes. This process is often precipitated by the chronic rubbing of or friction to the skin. Cytokeratins, which are the target of auto-antibodies, are released during the apoptosis of keratinocytes, and cytokeratin 5 (also 1, 10, and 14) is the major component of amyloid. After subsequent phagocytosis by macrophages, the resulting enzyme product becomes amyloid K. Clinically, MA presents as hyperpigmented, thin plaques, often with “wavy” linear grayish-brown streaks. The scapular region and extensor areas of the extremities are the most frequently affected areas. In contrast, LA is manifested as discrete, millimeter-sized skin-to-hyperpigmented colored dome-shaped papules. These papules overlap to form plaques, with the calves being a typical site. A rare variant known as “biphasic cutaneous amyloidosis” has been described, in which lichenoid and macular lesions occur at the same time [8].

Unlike the other two forms of primary localized amyloidosis, the amyloid in PLCNA is derived from immunoglobulin light chains (amyloid AL) (either κ or λ) and, thus, from plasma cells. However, an isolated case of PLCNA has been described in which the amyloid substance was found to be of the K-type rather than the AL-type [9]. Although, in systemic amyloidosis, the amyloid substance is also derived from immunoglobulin light chains, whereas in PLCNA, the plasma cells are found only in the skin and not in the bone marrow since it is not associated with an underlying hematologic dyscrasia. However, PLCNA could be considered a localized plasma cell dyscrasia, in which the cutaneous clusters of plasma cells act as an extramedullary plasmacytoma that produces amyloid fibrils [10]. Though occasional cases of plasma cell monoclonality have been described, current trends consider PNCLA to be a reactive rather than a neoplastic disease [5,10,11].

PLCNA manifests as single or multiple waxy, firm, pink, yellowish, or purplish plaques or nodules (Figure 1). Atypical presentations mimicking lymphatic malformation, milia, or bullous disease have also been reported [12,13,14]. PLCNA lesions are most commonly found on the acral region, head and neck, and extremities. The plaques and nodules may have a purpuric appearance or telangiectasias visible on their surface. Although they may be asymptomatic, pain is usually present, sometimes accompanied by itching. These symptoms are thought to be due to the mass effect on adjacent tissues [15,16]. Planas-Ciudad et al. [17] reported a case of PLCNA with dystrophic calcifications on ultrasonography and subsequent histologic confirmation. Dystrophic calcifications are rarely seen in PLCNA but have been described in cases of localized amyloidosis of other organs. Patients with PLCNA are sometimes reluctant to seek treatment and do so when lesions become symptomatic or for cosmetic reasons. Spontaneous local ulceration is not uncommon, and clinical changes may sometimes occur in response to local trauma. In contrast to MA and LA, for which it is well established that repeated local trauma from scratching leads to the formation of amyloid K, there are few articles suggesting the possible role of local trauma in the deposition of the immunoglobulin light chains involved in the genesis of PLCNA [18,19].

In PLCNA dermoscopy, the most common finding appears to be a yellowish background (without structure) interrupted by whitish, scar-like lines with some hemorrhagic areas and vessels resembling fine telangiectasias. This pattern may correspond to the nodular aggregates of amyloid in the reticular dermis and subcutaneous tissue, also involving the adnexa and vessel walls with a surrounding lymphoplasmacytic infiltrate. PLCNA should be considered in the differential diagnosis of yellow-orange lesions in dermoscopy. This group comprises several histiocytic and granulomatous diseases, both inflammatory, like xanthogranuloma or sarcoidosis, and infectious, such as leishmaniasis or lupus vulgaris. In PLCNA, the unstructured yellow aggregates are different from the small yellow dots of these granulomatous diseases, which have a more pronounced inflammatory infiltrate [20].

Histopathology is considered the gold standard for diagnosing PLCNA. The nodular deposition of hyaline and eosinophilic material is usually seen in the full thickness of the dermis. Such material may also be found in small vessel walls, adnexa, and subcutaneous tissue. Variable plasma cell infiltration may occur within amyloid deposits adjacent to the vasculature. Eosinophilic material corresponding to an amyloid substance is stained with Congo red, showing a characteristic brick-red deposit. Under polarized light microscopy, an apple-green birefringence is seen. The subtype of amyloid deposits must then be determined, which reveals the κ and/or λ light chain [20]. For diagnosis, in addition to biopsies, soft-tissue imaging, preferably magnetic resonance imaging (MRI), should be requested in cases of extensive or deep involvement [21].

Systemic amyloidosis is characterized by amyloid deposition in the internal organs (mainly myocardium, liver, and kidney), which can lead to progressive loss of function and, in advanced cases, even death. Patients with systemic amyloidosis may have skin involvement, although cutaneous involvement is rare in systemic amyloidosis secondary to inflammatory disease (amyloid AA). When it does occur, it is usually mild and presents as petechiae, purpura, and/or alopecia. Primary systemic amyloidosis is caused by the proliferation of monoclonal plasma cells (amyloid AL). The underlying disorders include entities such as multiple myeloma, Waldenström’s disease, or malignant lymphomas, among others. In the primary systemic form, skin lesions are common and can vary widely. Plaques and nodules with a waxy appearance and a yellowish-orange color similar to those described in PLCNA are frequent [22]. In contrast to PLCNA, lesions are more common in mucocutaneous areas, such as the orbits and nostrils [21]. In primary systemic amyloidosis, it is frequent to find petechial, hemorrhagic, and purpuric lesions due to dermal involvement. The ulceration of papulonodular lesions is not uncommon, and when the scalp is affected, it leads to cicatricial alopecia [22]. When the dermis is extensively infiltrated by amyloid, scleroderma-like skin changes (especially on the fingers) may occur, known as ‘scleroderma amyloidosum Gottron’. Blistering and nail dystrophy have also been reported [23]. As discussed, the amyloid in both this systemic form and PLCNA is of the AL type. The histopathology of the skin lesions will show virtually identical changes in both entities [15,16].

The risk of progression of systemic amyloidosis in PLCNA was initially thought to be close to 50% based on early case reports. However, from a series of a few patients followed over time, it has now been estimated to be around 1–7%, although the rate of paraproteinemia can be as high as 40% [4,24]. Although the development of the systemic form a posteriori is uncommon, regular long-term follow-ups should be performed to rule out the progression of systemic amyloidosis. A thorough physical examination and a blood test with a complete blood count and metabolic panel are recommended at the time of diagnosis. The evaluation of the presence of a monoclonal plasma cell population outside of skin lesions is achieved by electrophoresis and immunofixation in the serum and urine, where the absence of monoclonal protein (M) indicates PLCNA. Bone marrow examination must show the absence of monoclonal plasma cells. Some authors also suggest performing initial imaging studies, such as positron emission tomography (PET) scanning, to evaluate systemic involvement. In order to rule out cardiac involvement, an electrocardiogram may be ordered [21].

In summary, amyloidosis represents a heterogeneous group of diseases with a wide variety of ethology, course, prognosis, and treatment. In particular, PLCNA can have a clinical and pathologic presentation identical to that of systemic amyloidosis. Therefore, its recognition and subsequent screenings for systemic amyloidosis are essential.

### 2.2. Primary Localized Cutaneous Nodular Amyloidosis and Sjögren’s Syndrome: A Humoral Response-Based Relationship

SjS is a chronic auto-immune lympho-proliferative disease characterized by the inflammation and dysfunction of the secretory glands, mainly the salivary and lacrimal glands, resulting in a dry mouth and eyes. The exocrine glands are infiltrated by T and B cells and are progressively destroyed by the cellular and humoral response. The epithelial cells of the glands are currently considered to play an important role in the pathogenesis of SjS, as they are the triggers of immune activation. Thus, epithelial cells participate in this activation by expressing ribonucleoprotein complexes (i.e., Ro/SSa and La/SSb), interacting with T cells, and producing cytokines. Although T lymphocytes play an important role in pathogenesis, B lymphocytes are the main cells involved and are also implicated in SjS-associated non-Hodgkin’s B cell lymphoma. Increased levels of cytokines are normally associated with B lymphocytes, such as IL-6 and IL-10. In addition, serum B cell activating factor (BAFF) levels are elevated and germinal center-like structures have been identified in the salivary glands of patients with SjS. Ongoing B cell hyperactivity and light chain-producing plasma cells can lead to amyloid deposits in the skin [2]. The etiopathogenesis of PLCNA in patients with SjS is summarized in Figure 2.

SjS should be considered in any patient with PLCNA, as it is estimated that approximately one in four patients with PLCNA also has this disease [5]. In contrast, the true prevalence of PLCNA in patients with SjS is unknown. It is thought to be very low, although it may also be an underdiagnosed entity. Besides the skin, nodular amyloidosis in SjS has been sporadically described in the lungs [25] and breasts [26]. The plasma cells in the skin produce the immunoglobulin light chain that leads to the formation of fibril and, ultimately, to the deposition of amyloid. As the clonality of plasma cells has been reported, PLCNA could, therefore, be part of the spectrum of lympho-proliferative disorders associated with SjS [27]. Interestingly, in the management of this entity, immunosuppressive agents that interfere with B cell function do not appear to be useful. Indeed, Yong et al. [28] reported an immunosuppressed renal transplant patient with IgA nephropathy who developed PLCNA despite immunosuppressive therapy with azathioprine, prednisolone, and cyclosporine. Therefore, they postulated that systemic immunosuppressants are unlikely to affect local plasma cell secretion. In addition, Tong et al. [29] reported a case wherein a patient, who also failed multiple immunosuppressive therapies, finally responded to cyclophosphamide.

Reviewing the literature to date (see Table 1), 26 articles were found for a total of 34 patients with PLCNA and SjS. The mean age of the patients in the reported cases was 64.47 years, with a median of 63 years (35–88). Interestingly, all but one of the cases studied were in female patients (97%). In cases where it is specified (71.33%), the median time to the diagnosis of PLCNA was 4 years (6 months–26 years), with a mean of 5.57 years. The most common location of PLCNA in patients with SjS (Figure 3) was in the lower extremities (18/34), followed by the trunk (12/34). PLCNA manifested as multifocal in distinct body areas in 20.6% of the reported cases. There is a clear preponderance of Japanese patients in those articles where ethnicity is reported. A limitation of our analysis is that the available literature is based on retrospective articles (case series and case reports).

The marked female predominance of PLCNA cases in SjS is noteworthy, as previous series have shown no gender difference [4] or even a male predominance [24]. This disparity could be explained by the strong association of SjS with the female sex, as these series included cases regardless of whether they were associated with other diseases or not. Both series are retrospective with long-term patient follow-ups. The study by Woollons et al. [4] includes 15 cases. However, it is not indicated whether any of them also had SjS. The series by Moon et al. [24] includes 16 patients, two of whom also had SjS, but it is not specified which of these patients had this disease. Yoshida et al. [46] performed a comparative analysis based on cases reported in the literature in Japanese patients. The authors compared published epidemiologic data in Japanese patients with PLCNA with or without associated SjS. In patients with PLCNA and SjS, the F:M ratio was 5.6, whereas in patients without SjS, the F:M ratio was 0.55. Therefore, based on the few series of PLCNA reported in the literature, it can be assumed that this entity is equally prevalent in both sexes or slightly more prevalent in the male sex. However, in the presence of SjS, the ratio is clearly reversed. The explanation surely comes from the fact that SjS is found nine times more often in the female sex. PLCNA is most commonly seen in patients with or without associated SjS in the seventh decade of life.

The location of PLCNA is highly variable, with cases described in such odd locations as the areola or tongue. In general, acral and facial involvement is significantly more common in patients without SjS than in those with SjS. In contrast, in patients with SjS, the nodules are expected to be located either on the trunk or especially on the lower extremities. Only less than 20% of reported cases occur in locations other than these two. Therefore, there is greater variability in the location of lesions in patients without SjS.

The review of the literature also showed that the histopathology of PLCNA in the described cases did not vary when associated with SjS. The most common finding in these patients, as discussed above, was a homogeneous dermal nodular collection of eosinophilic material, which was positive under Congo red staining. The extension of the disease into the subcutaneous cellular tissue and the presence of dermal plasma cell infiltration does not appear to be uncommon.

The B cell hyperactivity present in SjS may justify its association with PLCNA. This type of amyloidosis may be part of the spectrum of lympho-proliferative disorders associated with SjS. The clinical peculiarities of PLCNA and the poor response to systemic treatments directed against B cells suggest that a significant part of its etiopathogenesis remains unknown.

### 2.3. Primary Localized Cutaneous Nodular Amyloidosis: A Therapeutic Challenge

Given the relative infrequency of the diagnosis and the scarcity of prospective studies, the management of PLCNA presents a special challenge. There are no clear management recommendations and no treatment of choice. Actually, the evidence for this topic comes from case reports. The most frequently reported therapies for PLCNA are aimed at removing or improving the appearance of the lesion. Among them, surgical removal is one of the most commonly used treatments. To date, therapies have included surgical excision, dermabrasion, electrodesiccation, and curettage. With these types of therapies, the results obtained are variable, and there is a high rate of recurrence for the lesions. In addition, it has not been effectively treated with other physical interventions, such as cryotherapy [50].

There have been good responses to laser treatment in PLCNA. Four case reports of CO2 laser treatment have been published. The first patient was a 60-year-old man with a 20-year history of disseminated non-Hodgkin’s lymphoma. Good improvement was noted, and the patient was followed for 12 months [51]. A patient with a 6-year history of nasal PLCNA after one treatment session and a patient with a lesion over the left temple after two treatment sessions also reported good improvements. In these cases, tissue fragility during treatment and atrophy were the complications reported [52,53]. There are also two published reports showing excellent responses to combined surgery and CO_2_ laser [54,55]. Pulsed dye laser (PDL) has been shown to be beneficial in the treatment of PLCNA.

Systemic treatment may be considered in extensive cases or when local modalities fail to control the disease [21]. As discussed, PLCNA often does not respond well to immunosuppressive therapies, although intra-lesional methotrexate may be an option for patients who have localized lesions and are poor candidates for surgery or invasive procedures [56]. In the case of Tong et al., a patient with CREST and SjS had a good response to cyclophosphamide. The patient showed skin ulceration resolution without further blistering and existing nodules stabilization. The use of cyclophosphamide in PLCNA is supported by its efficacy in myeloma and systemic AL amyloidosis [57]. Similarly, Khan et al. presented a case in which a good response to the combination of bortezomib and dexamethasone in a 64-year-old woman was observed. The patient had recurrent PLCNA and had previously undergone several local therapies with no response. The validity of this therapy was based on the good results obtained with the combination of proteasome inhibitors and corticosteroids in the treatment of systemic immunoglobulin light chain amyloidosis (AL) [21]. When taking into account the associated accompanying symptoms, interleukin-31 antibodies, as a novel antipruritic medication, may offer a new approach to the future treatment of this condition [50]. Further studies are needed to clearly identify the role of systemic therapy in PLCNA.

### 2.4. Primary Localized Cutaneous Nodular Amyloidosis: Just One of the Many Cutaneous Manifestations of Sjögren’s Syndrome

Like the mucous membranes, the skin is commonly affected in patients with SjS. Xeroderma, or skin dryness, is the most common dermatologic manifestation, being present in 67% of patients with primary SjS. It presents as rough and scaly skin, usually causing pruritus and a burning sensation. Eyelid eczema is a non-specific manifestation and is estimated to affect approximately 40% of patients. Cutaneous vasculitis occurs in approximately 15% of patients with primary SjS [58]. It can be a unique episode or recurring and most commonly presents as palpable or nonpalpable purpura on the lower extremities. Alternatively, other types of vasculitis may develop, such as urticarial vasculitis, which most commonly affects the face, upper extremities, and trunk. Unlike common urticaria, the lesions of urticarial vasculitis in SjS often persist for more than 24 h, are more painful than itchy, and may be accompanied by petechiae. Livedo reticularis is not rare in patients with SjS and may occur in the absence of vasculitis. Cutaneous vasculitis associated with SjS can be difficult to distinguish from benign Waldenström’s hyperglobulinemic purpura. The latter manifests with recurrent purpuric lesions, which are predominantly on the lower extremities, hemogram alterations, polyclonal hypergammaglobulinemia, and antibodies (to the Ro/SSa antigen). It is estimated that approximately 20% of patients with Waldenstrom’s hypergammaglobulinemic purpura have or will develop SjS or keratoconjunctivitis sicca [59,60].

In a series of 320 patients with SjS, García-Carrasco et al. observed that Raynaud’s phenomenon (RP) occurred in approximately 13% of patients with primary SjS. RP may also be an early manifestation, as it is estimated to be the first auto-immune symptom in almost half of the patients. The clinical course of RP appears to be milder in patients with primary SjS than in those with other systemic auto-immune diseases, such as systemic sclerosis [61]. In a low percentage of patients with SjS, a skin rash characterized by erythema annulare occurs. It manifests as polymorphic plaques with erythematous borders and a lighter center, similar to those seen in subacute lupus annulare. These lesions usually appear in light-exposed areas, heal without residual scarring, and are associated with positivity for anti-Ro/SSa and/or La/SSB antigens. A total of 80% of cases occur years after meeting the criteria for SjS [62]. As for lympho-proliferative disorders, some cases of lymphomatoid papillomatosis and cutaneous lymphomas associated with SjS have been described [63].

Some of the typical cutaneous manifestations of SjS may precede xerostomia and xerophthalmia. Therefore, the recognition of characteristic cutaneous signs is critical for the early diagnosis of SjS. Furthermore, the severity of systemic involvement and long-term prognosis may be correlated with different cutaneous findings. Patients with SjS and cutaneous vasculitis, especially cryoglobulinemic vasculitis, have been shown to have a worse prognosis. These patients would have an increased risk of systemic vasculitis and lymphoma. Waldenstrom’s hypergammaglobulinemic purpura has been associated with the development of peripheral sensory neuropathy [64]. Patients with RP may have an increased risk of extraglandular manifestations [58]. Those who develop photosensitive annular erythema appear to have a more indolent course with milder clinical signs at the glandular and systemic levels. With regard to PLCNA, it is not known what relationship it may have on the evolutionary course of the disease.

The treatment of cutaneous involvement in SjS should be individualized and will depend on the type of cutaneous manifestation. The frequent application of emollients may help to combat xerosis. Proper photoprotective habits are important given the relationship with other connective tissue diseases that may be exacerbated by photo-exposure, such as subacute lupus erythematosus. Topical corticosteroids are usually sufficient to treat uncomplicated vasculitis involving only the skin. Treatments such as dapsone, colchicine, systemic corticosteroids, and immunosuppressants may be used in more severe cases [65,66].

Cutaneous involvement is, therefore, very common in patients with SjS. The skin manifestations that can occur in these patients are many and varied. The detection and recognition of those with prognostic implications is important in the multidisciplinary management of these patients.

### 2.5. Primary Localized Cutaneous Nodular Amyloidosis and Other Auto-Immune Connective Tissue Diseases

The majority of case reports documenting PLCNA in association with autoimmunity have been in the setting of SjS. PLCNA has also been described in association with other connective tissue diseases such as CREST syndrome (calcinosis, Raynaud’s syndrome, esophageal involvement, sclerodactyly, and telangiectasia) [67,68], systemic sclerosis [20,69], and systemic lupus erythematosus [70]. This suggests that the dysregulation of plasma cells in the skin may be related to the auto-immune imbalance that is inherent in connective tissue diseases. Limited systemic sclerosis (including CREST syndrome) appears to be the entity most commonly associated with PLCNA after SjS. As with the latter, PLCNA in limited systemic sclerosis is more common in postmenopausal females and in the lower extremities. Atzori et al. suggested that the progressive hardening of soft tissues, depletion of appendages, and macro- and microvascular involvement that occur in scleroderma, particularly in the distal extremities, may contribute to the isolation of a pool of monoclonal plasma cells in the dermis. The light chains released by these cells would accumulate and could not be cleared due to poor circulation. The microtraumas of these sites are additional possible causes of PLCNA, including micro-calcifications [20]. Shiman et al. also proposed that the dysregulation of growth factors and cytokines that occurs in limited systemic sclerosis, as well as in other auto-immune diseases, may, in part, account for the proliferation of extramedullary plasma cells [68].

## 3. Future Perspectives

The authors believe that it is important to continue the trend of recent years, in which an increasing number of cases of PLCNA have been described in patients with SjS. This may help to better define and understand this special relationship. In addition, it could help to know if the development of PLCNA is a prognostic factor in the evolution of SjS. Finally, prospective studies to evaluate therapeutic options in PLCNA would be useful, although this is complicated by the paucity of reported clinical cases.

## 4. Conclusions

PLCNA is a very rare disease that is closely related to SjS. Therefore, if a diagnosis of PLCNA is confirmed, the physician should rule out associated SjS as well as other connective tissue diseases and screen for systemic amyloidosis. After reviewing the literature, it was observed that the co-existence of PLCNA and SjS has been described mainly in female patients in their seventh decade of life with nodular lesions on the trunk and/or lower extremities. Acral and facial localization, which is a typical localization of PLCNA, seems to be much more unusual in patients with associated SjS.

## Figures and Tables

**Figure 1 ijms-24-07378-f001:**
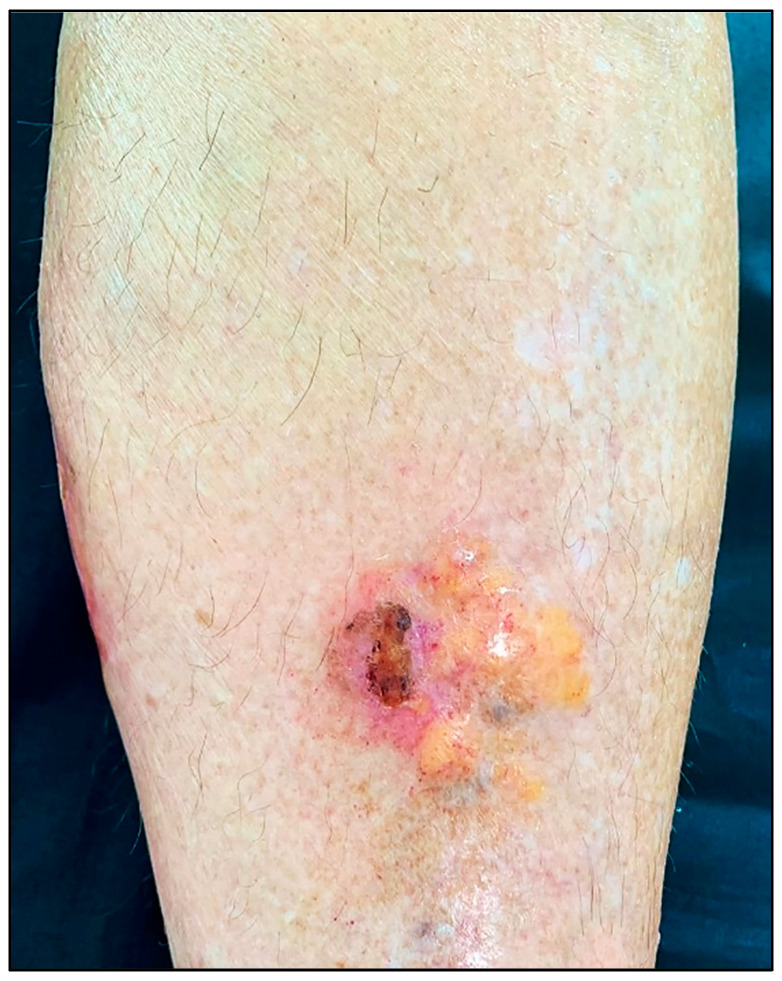
Patient with primary localized cutaneous nodular amyloidosis (PLCNA). Partially ulcerated yellowish nodule on the anterior aspect of the left leg.

**Figure 2 ijms-24-07378-f002:**
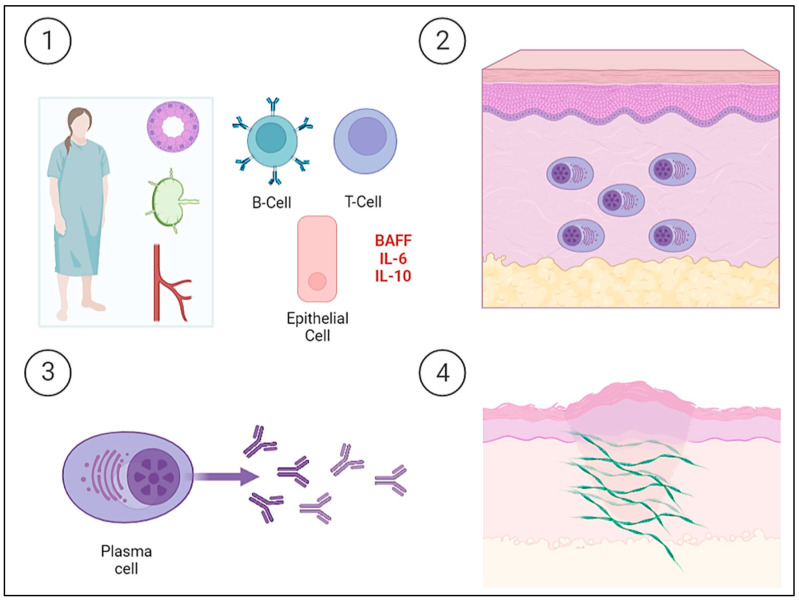
Schematic representation of the etiopathogenesis of Sjögren’s Syndrome (SjS) and the development of primary localized cutaneous nodular amyloidosis (PLCNA). ① The epithelial cells of the salivary gland itself and acquired immunity both play a key role in the etiopathogenesis of a patient with SjS. Epithelial cells act as initiators of the immune response by expressing ribonucleoproteins on their surface, producing cytokines, and interacting with T lymphocytes. Tissue infiltration and progressive destruction by T and B cells occur, with the latter being the fundamental cells in the etiopathogenesis and producing cytokines, such as interleukin-6 (IL-6) and interleukin-10 (IL-10). There is also an increase in serum B cell activating factor (BAFF) levels. ②, ③ B cell hyperactivity leads to the deposition of light chain-producing plasma cells in the skin. ④ The deposition of light chains leads to the formation of AL amyloid protein. The cutaneous “amyloidoma” gives rise to the characteristic plaques and nodules with a tendency to become chronic and ulcerated.

**Figure 3 ijms-24-07378-f003:**
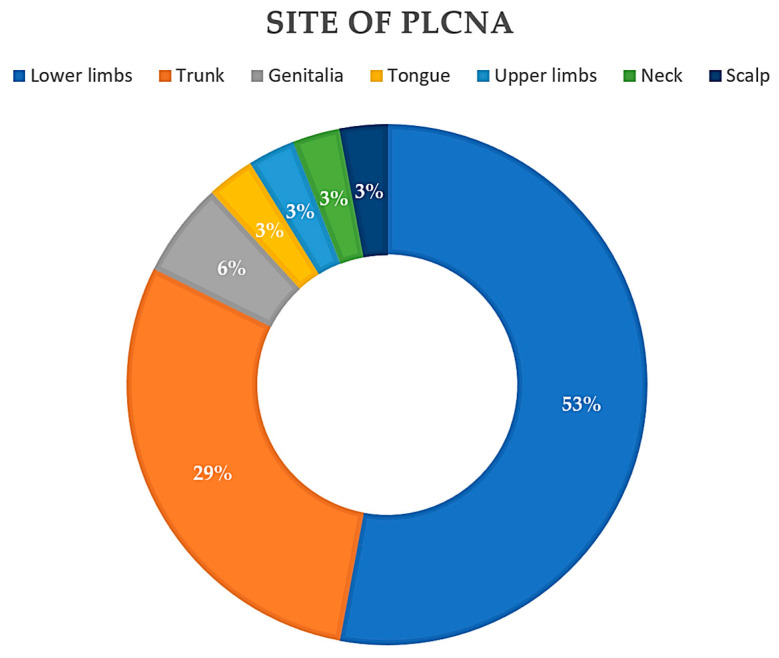
Pie chart. Reported locations of primary localized cutaneous nodular amyloidosis (PLCNA) in the 34 patients described to date with Sjögren Syndrome (SjS).

**Table 1 ijms-24-07378-t001:** Cases described in the literature for patients with Sjögren’s syndrome and PLCNA.

Case	Age	Sex	Ethnic Group	Site of PLCNA	PLCNA Duration	SjS Duration before PLCNA	Histopathological Findings	Congo Red	Ref.
1	63	F	Japanese	Trunk and lower extremities	10 years	3 years	Hyaline eosinophilic masses almost the entire dermis	+	[30]
2	53	F	Not reported	Genitocrural region	4 years	Not reported	Eosinophilic amorphous material	+	[31]
3	35	F	Not reported	Neck	4 years	18 years	A mass of amorphous eosinophilic material with numerous plasma cells surrounded	+	[32]
4	57	F	Japanese	Trunk	Several months	1 year	Amorphous and eosinophilic masses throughout the entire dermis.	+	[33]
5	68	F	Not reported	Right shin	3 years	5 years	Abundant hyaline eosinophilic masses almost the entire dermis.	+	[34]
6	70	F	Japanese	On the bilateral calves of legs	3 years	1 year	Amyloid protein deposition in almost the entire dermis.	+	[35]
7	51	F	Japanese	On the hip	1 year	1 year	Amyloid protein deposition in almost the entire dermis.	+	[35]
8	53	F	Japanese	On the left waist, left thigh, and right lower leg	Not reported	18 years	Massive amorphous eosinophilic deposits in the reticular dermis	+	[36]
9	77	M	Not reported	Right axilla and left chest	1 year	17 years	Not reported	Not reported	[37]
10	62	F	Not reported	Tongue	Not reported	5 years	Amorphous and eosinophilic masses	+	[38]
11	69	F	Not reported	Trunk	10 years	7 years	Hyaline eosinophilic masses almost the entire dermis	+	[39]
12	71	F	Hispanic	Scalp	6 months	Not reported	Homogeneous eosinophilic material throughout the reticular dermis	Not reported	[40]
13	69	F	Japanese	Genitalia	2 years	6 years	Eosinophilic material through the entire dermis into the subcutaneous fatty tissue. The deposited material was also found within the walls of blood vessels	+	[41]
14	71	F	Not reported	Left leg	14 years	4 years	Not reported	+	[5]
15	50	F	Not reported	Arm, back, legs	5 years	3 years	Not reported	+	[5]
16	69	F	Not reported	Trunk	26 years	8 years	Not reported	+	[5]
17	61	F	Not reported	Back	19 years	14 years	Not reported	+	[5]
18	88	F	Not reported	Legs	9 years	43 years	Not reported	+	[5]
19	83	F	Not reported	Legs	4 years	52 years	Not reported	+	[5]
20	63	F	Not reported	Arm, legs, trunk	4 years	33 years	Not reported	+	[5]
21	63	F	Not reported	Trunk	2 years	2 years	Not reported	+	[5]
22	52	F	Not reported	Legs	Not reported	20 years	Dermal nodular homogeneous collection of an eosinophilic material	+	[16]
23	78	F	Not reported	Legs	4 years	4 years	Homogeneous eosinophilic material more prominent surrounding blood vessels and appendages and perivascular plasma cell infiltration	+	[42]
24	56	F	Not reported	Feet and calves bilaterally	18 months	Not reported	Abundant deposits of eosinophilic homogenous material within blood vessels as well as surrounding adipocytes	+	[29]
25	41	F	Not reported	Abdomen, back, and buttocks	3 months	11 years	Homogenous, amorphous, and basophilic material diffusely throughout the dermis	+	[27]
26	78	F	Not reported	Anterior aspect of the left leg	7 years	17 years	Eosinophilic dermal-hypodermal deposits	+	[43]
27	60	F	Not reported	Breast and bilateral upper extremities	Not reported	Not reported	Amorphous, eosinophilic deposits within the dermis	+	[44]
28	59	F	Not reported	Genitocrural region	Several years	20 years	A mass of amorphous eosinophilic material with numerous plasma cells surrounded	+	[6]
29	78	F	Chinese	Legs	3 years	10 years	Amorphous and eosinophilic masses throughout the dermis	+	[45]
30	88	F	Japanese	Back and chest	6 months	Not reported	Monomorphous materials throughout the dermis, surrounded by lymphocytes and plasmacytes	+	[46]
31	55	F	Caucasian	Left shin	1 year	8 years	Extensive stromal deposition of eosinophilic material	+	[7]
32	73	F	Japanese	Legs	Several years	Not reported	Eosinophilic amorphous material and the infiltration of plasma cells	+	[47]
33	61	F	Not reported	Nape, upper back and left cheek	Not reported	Not reported	Amorphous eosinophilic deposition in the dermis	+	[48]
34	67	F	Japanese	Legs	6 years	Not reported	Diffuse deposition of an eosinophilic material through the entire dermis and subcutaneous tissue and a plasma cell infiltrate	Not reported	[49]

## Data Availability

No new data were created or analyzed in this study. Data sharing is not applicable to this article.

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
