# Peer review of "Localized Cutaneous Nodular Amyloidosis: A Specific Cutaneous Manifestation of Sjögren’s Syndrome"

_ijms, 2023, doi:10.3390/ijms24087378_

Round 1

Reviewer 1 Report

In the paper, the authors discuss about primary localized cutaneous nodular amyloidosis (PLCNA), a rare condition attributed to plasma cell proliferation and deposition of immunoglobulin light chains in the skin. Patients diagnosed with PLCNA may also suffer from other autoimmune connective tissue diseases, with Sjögren's syndrome (SjS) showing the strongest association. The authors provide a literature review and descriptive analysis highlighting the relationship between these two entities.

The review is interesting and well-written, however some improvements/additions may increase the range of potential readers and the overall diffusion of this review among the scientific community.

 -Paragraph “2.5. Primary Localized Cutaneous Nodular Amyloidosis and other autoimmune connective tissue diseases”. We believe that this paragraph is of much interest for the scientific community and physicians, in particular. The authors should improve this paragraph.

-“SjS should be considered in any patient with PLCNA, as it is estimated that approximately 1 in 4 patients with PLCNA also have this disease [4].” On the other hand, how many patients with SjS have PLCNA?

As a general point, we suggest to improve the epidemiological data reported in the study.

 -“The marked female predominance of PLCNA cases in SjS is noteworthy, as previous series have shown no gender difference [24] or even a male predominance [25]. This disparity could be explained by the strong association of SjS with female sex, as these series included cases regardless of whether they were associated with other diseases or not. Both series are retrospective with long-term follow-up of patients.”

We suggest to better discuss about this point. What is the female-to-male ratio for PLCNA, when this disease is diagnosed alone or in association with SjS and other autoimmune disorders?

Reviewer 2 Report

The authors summarize the evidence for the association between primary localized cutaneous nodular amyloidosis (PLCNA) and Sjögren's syndrome (SjS). They list prior studies on the PLCNA as a small subset of the large group of cutaneous amyloidosis, the humoral response-based relationship between PLCNA and SjS, the therapeutic challenge of PLCNA, other manifestations of SjS, and the association between PLCNA and other autoimmune connective tissue diseases. The authors provided one interesting figure that summarizes the reported locations of PNCLA in the 34 patients described to date with SjS. They also provided one table that characterized the clinical cases from the literature of patients with Sjögren's syndrome and PLCNA. From this discussion, the authors concluded that the coexistence of PLCNA and SjS has been described mainly in female patients in the seventh decade of life with nodular lesions on the trunk and/or lower extremities. Acral and facial localization, which is a typical localization of PLCNA, seems to be much more unusual in patients with associated SjS.

The current information is interesting, and the review is well-written.

Comments:    

1) The authors addressed each section adequately by providing narration for the evidence of discussed points. However, the authors still need to add a take-home message describing the critical aspects/reflection points at the end of each section.

2) In order to attract the interest of more readers and to make it clearer for them, the authors are advised to add an additional colored figure that provides a mechanistic schematic summary of section 2.2. (2.2. Primary Localized Cutaneous Nodular Amyloidosis and Sjögren's Syndrome: A humoral response-based relationship).

3) To make it clearer and more informative for readers and junior clinicians, providing a separate figure with a compilation of PLCNA pics from previous literature and histopathology – after obtaining a copyright permit – would be very interesting.

4) The authors are advised to describe the plan for discussing the current review in the past tense rather than the future tense “will”.

For example, in the introduction section (page 2, 3rd paragraph), the authors state “In addition, a descriptive analysis will be performed by collecting all cases of PLCNA in patients with SjS described in the literature”.

5) The authors are advised to make the legends of the figure/table stand alone. To this end, authors should provide the full names of all the listed abbreviations in the figure/table, including PLCNA.

Reviewer 3 Report

The presented review paper focuses on the specific cutaneous manifestation of Sjögren's Syndrome - Cutaneous Nodular Amyloidosis: (CNA)

The discussed topic is interesting and still gaining more and more attention in the field of immunology and dermatology, I like the structure, and the flow of the draft, however, I have some comments.

In general, I have two very major comments, that must be addressed and fixed before processing further.

Please follow my two points.

Firstly, the aim of the review paper is to discuss in a comprehensive manner the recent advances in the discussed area. Here, I see lots of references dated 2005-2015. Recently, basic databases like W-O-S or PubMed shows over 300 records for similar searches. Only a few are included in the review. The novelty and very narrow literature search is the major problem that must be fixed.

Moving further (even for narrative reviews), there is a very limited amount of information given regarding the following PRISMA guidelines and description of the searching strategy, which is so crucial for review papers. The Author should include at least: Data sources and searches, Study eligibility criteria, Study selection process, Data extraction, and study quality assessment (assessing the risk of bias (ROB) for each included study), Data synthesis. MeSH terms (in addition/replacement of keywords) are necessary to be included. For each step, it is necessary to explain to the reader with pictures or tables. It is necessary to explain what was drawn at each step to lead to the result. Moreover, a figure showing the PRISMA-based workflow must be drawn accordingly to the Prisma schema. After that, a discussion is valuable even for narrative papers. A description of the Data Mining strategy should also be included.

I would also suggest creating a figure which will bring some new mechanistic approaches to the discussed area - the presence of a good figure sometimes says more than 1000 words.

Please unify references citation pattern - e.g. ref #67, #68 #69 etc etc

There are some minor grammatical errors / minor typos - please go over them carefully.

Round 2

Reviewer 1 Report

issues addressed